# Fluctuation Effects of Magnetohydrodynamic Micro-Vortices on Odd Chirality in Magnetoelectrolysis

**Iwao Mogi [1],\*, Ryoichi Aogaki [2] and Kohki Takahashi [1]**

[1] Institute for Materials Research, Tohoku University, Katahira, Aoba-ku, Sendai 980-8577, Japan; kohki@imr.tohoku.ac.jp

[2] Department of Product Design, Polytechnic University, Sumida, Tokyo 130-0026, Japan; AOGAKI.Ryoichi@nims.go.jp

\* Correspondence: mogi@imr.tohoku.ac.jp

**Abstract:** The magnetic field dependence of chiral surface formation was investigated in magnetoelectrodeposition (MED) and magnetoelectrochemical etching (MEE) of copper films. The MED and MEE was conducted in magnetic fields of up to 5 T, which were parallel or antiparallel to the ionic currents. The MED films prepared in high magnetic fields of 5 and 3 T exhibited odd chirality for magnetic field polarity, as expected on the basis of the magnetohydrodynamic (MHD) vortex model. However, the films prepared in the lower fields of 2.5 and 2 T exhibited breaking of odd chirality. Similar magnetic field dependence was observed in the surface chirality of MEE films. These results imply that the fluctuation in the self-organized state of micro-MHD vortices is responsible for the breaking of odd chirality.

**Keywords:** surface chirality; magnetoelectrodeposition; magneto-electrochemical etching; alanine; micro-MHD vortex; chiral symmetry breaking

---

## 1. Introduction

In biomolecular systems, L-amino acids are active, whereas D-amino acids are inactive. Such chiral symmetry breaking is well known; however, the origin of homochirality is one of the most attractive mysteries in natural science. In the molecular evolution towards the birth of life, surfaces of minerals could serve as catalysts in the formation of amino acids under the oceans on the early earth [1,2]. If such catalytic surfaces were chiral, the chiral symmetry would be broken between the enantiomers of amino acids. Thus, chiral surface formation and its chiral symmetry are crucial points for the origin of homochirality of amino acids.

We have demonstrated that magnetoelectrodeposition (MED), representing electrodeposition under magnetic fields, produces chiral surfaces on metal thin films [3–7]. The MED films exhibited chiral recognition for the enantiomers of amino acids and glucose. In MED processes with magnetic fields perpendicular to a working electrode, the Lorentz force acting on ionic currents $i$ induces two types of MHD flow [8,9], as shown in Figure 1a. The first one is a vertical MHD flow, which is a macroscopic flow around the electrode edge. The second one consists of micro-MHD vortices, which arise locally around bumps and pits generated by the non-equilibrium fluctuation. The micro-MHD vortices could contribute to the chiral surface formation through the screw dislocation growth [10]. The micro-MHD state has both clockwise and anticlockwise vortices such that the adjoining vortices never conflict with each other. Such a symmetrical state can be broken by the vertical MHD flow. The cyclonic vortices are stable, whereas anticyclonic ones become unstable [11,12]. The stable micro-MHD vortices could contribute to the screw dislocation growth. The magnetic field polarity determines the direction of vertical MHD flow. Hence, the chiral sign of the film surface becomes opposite when the magnetic

field is reversed, representing odd chirality for magnetic field polarity [13]. The odd chiral behaviors were obviously observed in the MED of copper films in a magnetic field of 5 T [4,5], where the rigid self-organized MHD state shown in Figure 1a could be formed.

　　We have also demonstrated that magnetoelectrochemical etching (MEE) produces chiral surfaces on copper films and the MEE films exhibit odd chirality for the magnetic field polarity at 5 T [14,15]. This indicates the formation of self-organized micro-MHD states during the MEE processes as well as the MED ones.

　　On the other hand, we have found the breaking of odd chirality in the copper MED with chloride additives, where the MED films showed only L-active chirality in both magnetic field polarities [16]. Moreover, the chloride additives cause similar broken odd chirality in the MEE of copper films [15]. These findings are surprising and of great interest in connection with the origin of homochirality in biomolecules. We have explored the experimental conditions to cause the breaking of odd chirality in copper MED. The effects of chloride additives suggest that the specific adsorption of chloride ions on the film surfaces induces substantial fluctuation in the self-organized MHD state, as shown in Figure 1b, and such fluctuation might be essential for the breaking of odd chirality. The fluctuation in the micro-MHD vortices can be expected when the magnetic fields are not strong enough to form the self-organized MHD state. In this paper, we report the chiral behaviors of the copper films prepared by the MED and MEE with the relatively low magnetic fields of 1~3 T.

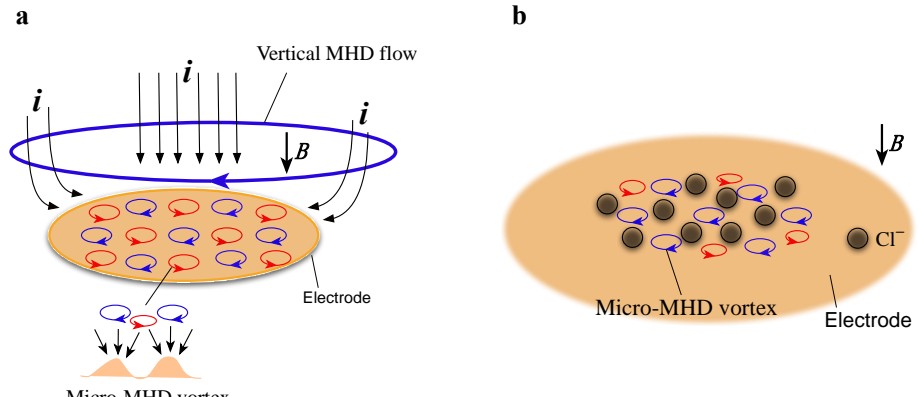

**Figure 1.** (**a**) Self-organized state of micro-MHD vortices under the vertical MHD flow. (**b**) Fluctuation of micro-MHD vortices in the presence of specific adsorption of chloride ions.

## 2. Results and Discussion

### 2.1. Effects of Low Magnetic Fields on MED

　　The surface chirality of MED films was examined by the measurements of voltammograms of alanine enantiomers on the MED film electrodes. Figure 2a shows voltammograms of L- and D-alanines on the MED +5T-film electrode, which was prepared at a deposition current of 20 mA cm$^{-2}$. The current peaks around 0.7 V correspond to the oxidation reaction of alanine on a Cu film electrode [17]. The chiral behavior is seen in the peak current difference between the enantiomers. The peak current of L-alanine is greater than that of D-alanine. Such chiral behavior is termed L-activity. On the contrary, the MED−5T-film electrode prepared at 17 mA cm$^{-2}$ exhibits D-activity (Figure 2b).

　　We examined the chiral behavior of the MED film electrodes prepared at various deposition currents. To evaluate the chirality in voltammograms, an enantiomeric excess (*ee*) ratio was defined as

$$ee = (i_p{}^L - i_p{}^D)/(i_p{}^L + i_p{}^D) \qquad (1)$$

where $i_p{}^L$ and $i_p{}^D$ represent the peak currents of L- and D-alanines, respectively. The positive sign of the *ee* ratio stands for L-activity, and the negative one stands for D-activity. Figure 3a,b show the

*ee* ratios versus the deposition currents (*ee* ratio profile) for the +5T-film and −5T-film electrodes, respectively. The +5T-film shows D-activity in the current region around −11 mA cm$^{-2}$ and then L-activity around −20 mA cm$^{-2}$. On the contrary, the −5T-film shows L-activity around −8 mA cm$^{-2}$ and D-activity around −17 mA cm$^{-2}$ (Figure 3b). These results indicate that the chiral sign depends on the deposition current and the magnetic field polarity.

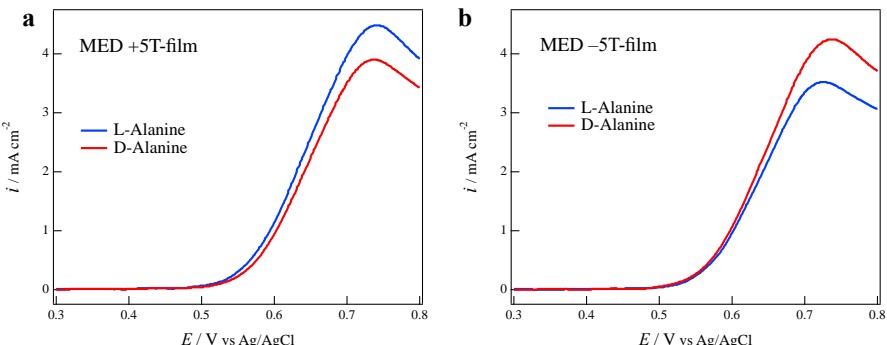

**Figure 2.** Voltammograms of alanine enantiomers on the MED Cu film electrodes: (**a**) on the +5T-film electrode prepared at a deposition current of 20 mA cm$^{-2}$; (**b**) on the −5T-film electrode prepared at 17 mA cm$^{-2}$.

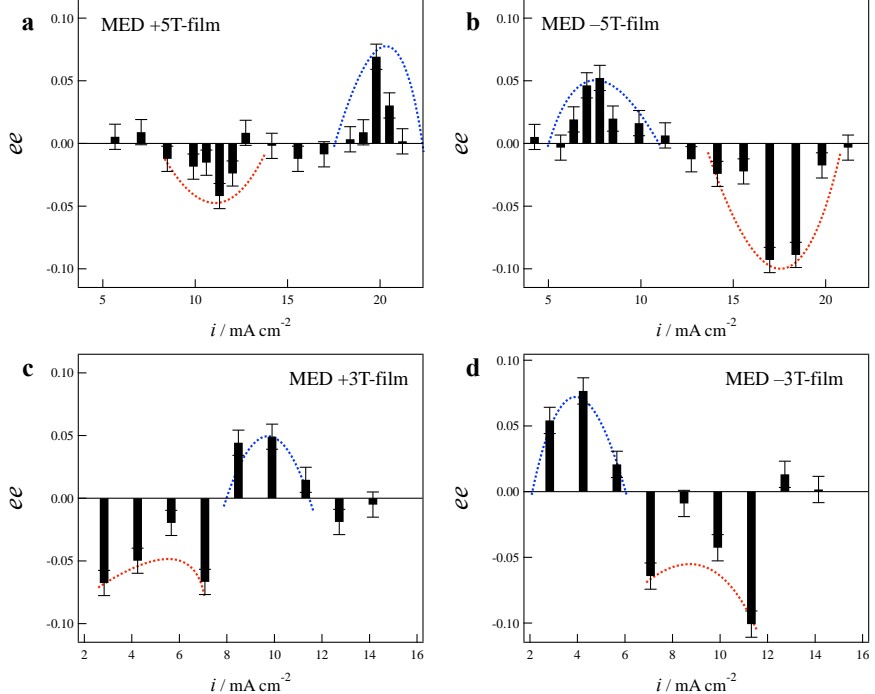

**Figure 3.** The *ee* ratios vs. deposition currents. (**a**) MED +5T-film, (**b**) MED −5T-film, (**c**) MED +3T-film and (**d**) MED −3T-film. Dotted curves are a guide for the eye.

Our previous paper reported the current dependence of chirality of the MED Cu films and indicated that the rate-limiting step in electrodeposition is responsible for the surface chirality of MED films [16]. The chirality could be different between kinetic-controlled and mass-transfer-controlled processes. On the other hand, the magnetic field polarity dependence can be easily understood by the self-organized MHD model in Figure 1a. The sign of *ee* ratio becomes opposite when the magnetic field polarity is reversed. This can be described as

$$ee(\boldsymbol{B}) \approx -ee(-\boldsymbol{B}). \tag{2}$$

The rigid self-organized MHD state causes the odd chirality for magnetic field polarity, as described above.

Similar results are seen in the *ee* profiles of 3T-films, as shown in Figure 3c,d. The +3T-film exhibits D-activity in the lower current region (−2.5~−7.5 mA cm$^{-2}$) and L-activity in the higher region (−8~−11 mA cm$^{-2}$). On the contrary, the −3T-film exhibits L-activity in the lower current region and D-activity in the higher region. The 3T-films also represent the odd chirality for magnetic field polarity, and the results in Figure 3 indicate that the rigid self-organized MHD state is formed in the MED at 5 and 3 T.

As the imposed magnetic field is decreased, the formation of a rigid self-organized MHD state would be difficult; then, the change in *ee* ratio profile is of great interest. Figure 4 shows the *ee* ratio profiles of the MED Cu films prepared at 2.5 and 2 T. The +2.5T-film exhibits L-activity around 7 mA cm$^{-2}$ (Figure 4a), and the −2.5T-film also exhibits L-activity in the same current region (Figure 4b). In the case of 2T-films, the +2T-film exhibits D-activity around 8 mA cm$^{-2}$ and L-activity in the region of 10~12 mA cm$^{-2}$ (Figure 4c). The −2T-film also exhibits almost the same behavior (Figure 4d). Both 2.5T- and 2T-films show the same chiral sign in both magnetic field polarities, being described as

$$ee(\boldsymbol{B}) \approx ee(\text{-}\boldsymbol{B}). \tag{3}$$

This relation represents the breaking of odd chirality, namely the even chirality for the magnetic field polarity. The even chirality is symmetry-forbidden in the MHD model, and it can be allowed in the gradient force fields. However, the electrochemical cell was placed in the homogeneous magnetic fields at the center of magnet bore; thereby, there is no gradient force. These facts imply that the breaking of odd chirality could take place in low magnetic fields such that the self-organized MHD state could not be formed rigidly.

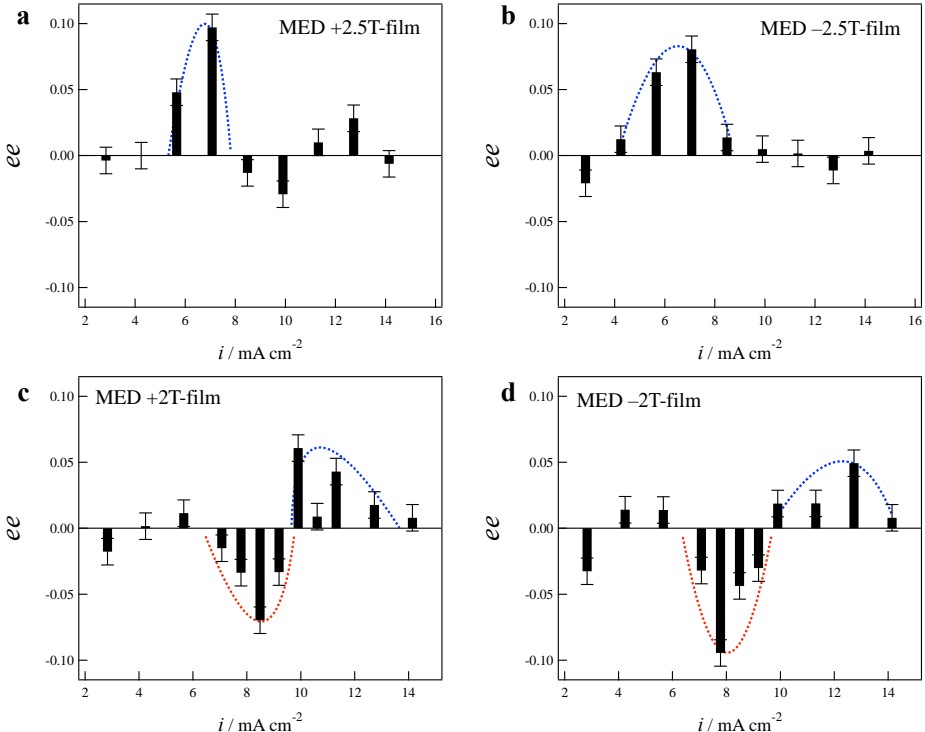

**Figure 4.** The *ee* ratios vs. deposition currents. (**a**) MED +2.5T-film, (**b**) MED −2.5T-film, (**c**) MED +2T-film and (**d**) MED −2T-film. Dotted curves are a guide for the eye.

The *ee* ratio profiles of 0T- and 1T-films were examined, and the chiral behavior disappeared in both films. The chiral surface cannot be formed when the micro-MHD vortices are absent or not fully developed.

## 2.2. Effects of Low Magnetic Fields on MEE

The MEE processes can also produce chiral surfaces [14]. Figure 5a,b show the *ee* ratio profiles for the MEE 5T-films. The +5T-film shows D-activity in the current region of 20–30 mA cm$^{-2}$ (Figure 5a), whereas the −5T-film shows L-activity in the same current region (Figure 5b), representing clear odd chirality for the magnetic field polarity. The 3.5T-films exhibit similar odd chirality in the low current region around 5–10 mA cm$^{-2}$ (Figure 5c,d).

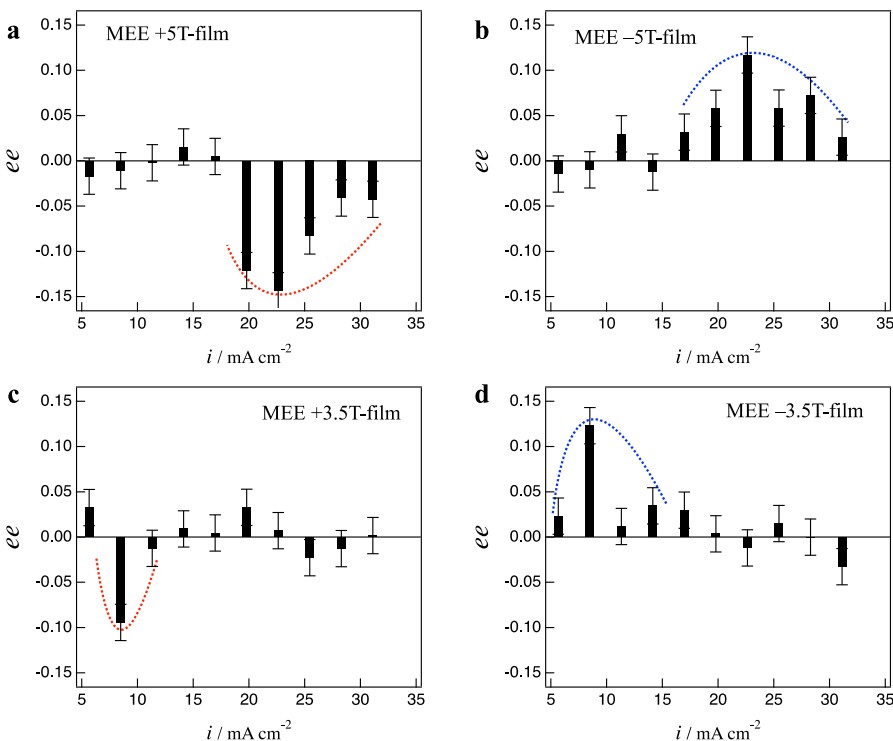

**Figure 5.** The *ee* ratios vs. etching currents. (**a**) MEE +5T-film, (**b**) MEE −5T-film, (**c**) MEE +3.5T-film and (**d**) MEE −3.5T-film. Dotted curves are a guide for the eye.

The chiral behaviors of MEE films prepared in the lower magnetic fields are substantially different. Figure 6 shows the *ee* ratio profiles for the 3T- and 2.5T-films. The +3T-films exhibit D-activity in the whole current region (Figure 6a), and the −3T-films exhibit the same profile (Figure 6b). On the other hand, the +2.5- and −2.5-films exhibit L-activity in the whole current region. The odd chirality is clearly broken in both 3T- and 2.5T-films. These results imply that the fluctuation of the micro-MHD vortices in low magnetic fields is responsible for the breaking of odd chirality. As well as the MED films, the 1T- and 0T-films showed the disappearance of chiral behavior.

## 2.3. Superimposed Effects of Low Magnetic Fields and Specific Adsorption

To examine the influence of more random fluctuation on chiral surface formation, we conducted the MED and MEE in low magnetic fields with the specific adsorption of chloride ions. Figure 7 shows the *ee* ratio profiles of the MED and MEE films prepared in the CuSO$_4$ + H$_2$SO$_4$ electrolytic solution containing 0.2 mM KCl. Both MED +2.5T- and −2.5T-films have small *ee* ratios of less than ±0.03 and exhibit no chiral tendency in the whole current region (Figure 7a,b); thus, they are regarded as achiral films. Similarly, the MEE +3T- and −3T-films representing tiny *ee* ratios are achiral (Figure 7c,d).

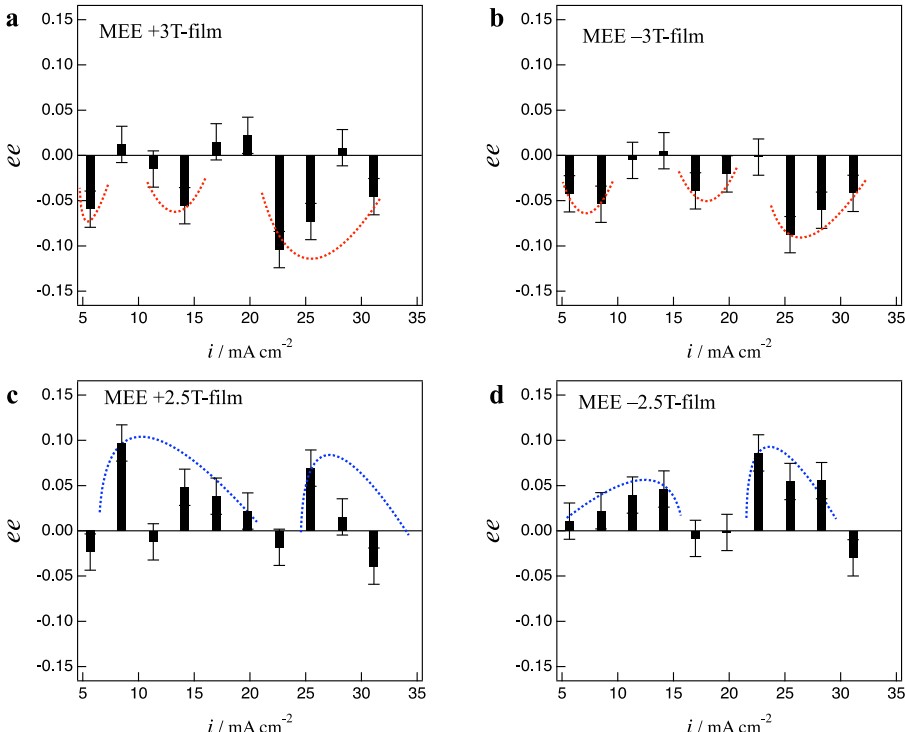

**Figure 6.** The *ee* ratios vs. etching currents. (**a**) MEE +3T-film, (**b**) MEE −3T-film, (**c**) MEE +2.5T-film and (**d**) MEE −2.5T-film. Dotted curves are a guide for the eye.

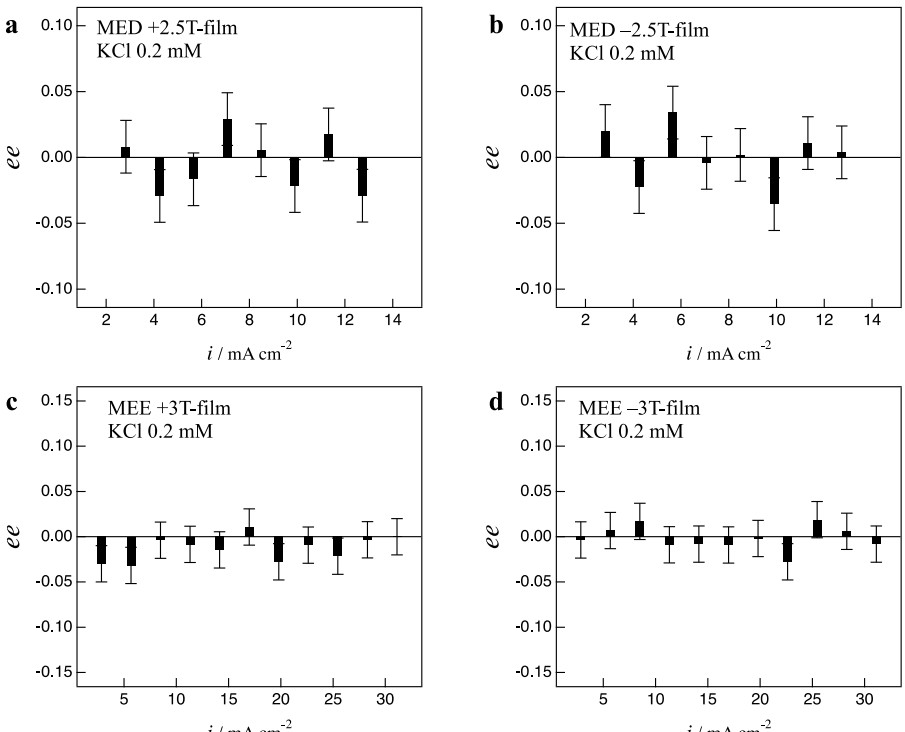

**Figure 7.** The *ee* ratio profiles of the MED and MEE films prepared with KCl additives. (**a**) MED +2.5T-film, (**b**) MEE −2.5T-film, (**c**) MEE +3T-film and (**d**) MEE −3T-film.

As reported in our previous papers [15,16], the specific adsorption of chloride ions causes fluctuation in the self-organized micro-MHD vortices. As mentioned above, the low magnetic fields of 2~3 T also cause fluctuation in the micro-MHD vortices. Both effects lead to the breaking of odd chirality

in the MED and MEE films. However, the superimposition of both effects results in the disappearance of chirality on the film surfaces, as shown in Figure 7. This indicates that such superimposition could bring about more random fluctuation in the micro-MHD vortices and disturb the self-organized MHD states, resulting in the disappearance of surface chirality.

On the other hand, in the case of the broken odd chirality in Figures 4 and 6, the surface chirality actually exists in the MED and MEE films. Thus, the self-organized MHD state could be formed with the fluctuation of micro-MHD vortices. Such fluctuation is not too random to disturb the self-organization of micro-MHD vortices. There exists "ordered fluctuation" in the micro-MHD state, where the correlation among fluctuations survives through the hydrodynamic interactions among the micro-MHD vortices. The micro-MHD states with ordered fluctuation produce chiral surfaces, and their chiral signs could be independent of the magnetic field polarity. The latter might be related to the correlation among fluctuations, but the detailed mechanism remains unsettled.

Similar broken odd chirality was observed in the MED with micro-electrodes (~100 μm) [18] and the rotational MED [19]. In the former case, the strong vertical MHD flows around the micro-electrodes cause fluctuation in the micro-MHD vortices. In the latter, the cell rotation in magnetic fields induces fluctuation in the self-organized MHD state through the precession of micro-MHD vortices. In both cases, the MED films exhibit clear surface chirality; hence, the fluctuation is not so random that the ordered fluctuation could survive. A number of experimental results so far suggest that the ordered fluctuation in the micro-MHD vortices is crucial for the breaking of odd chirality. The theoretical description of the ordered fluctuation is a future issue and it would lead to a deeper understanding of chiral symmetry breaking.

## 3. Materials and Methods

### 3.1. Electrodeposition and Electrochemical Etching

The electrochemical cell in both MED and MEE processes consists of three conventional electrodes: a polycrystalline Pt disc working electrode with a diameter of 3 mm, a Cu plate counter electrode and a Ag|AgCl|3 M (M = mol dm$^{-3}$) NaCl reference electrode. The Cu films were electrodeposited on the working electrode by the galvanostatic electrodeposition at constant currents of 2–22 mA cm$^{-2}$ in a 50 mM CuSO$_4$ aqueous solution containing 0.5 M H$_2$SO$_4$. The total passing charge was 0.4 C cm$^{-2}$ in all the MED experiments, and the film thickness was approximately 150 nm.

As for the etching procedures, Cu films were prepared first with a thickness of approximately 300 nm on the working electrode by electrodeposition in a 50 mM CuSO$_4$ + 0.5 M H$_2$SO$_4$ aqueous solution in the absence of a magnetic field. The electrochemical etching of the Cu films was conducted in the CuSO$_4$ + H$_2$SO$_4$ solution in galvanostatic conditions with various constant currents of 3.0–32 mA cm$^{-2}$. The passing charges were 0.80 C cm$^{-2}$ in the electrodeposition and 0.40 C cm$^{-2}$ in the MEE process, and the MEE film thickness was approximately 150 nm.

To examine the effects of chloride additives on the surface chirality, the MED and MEE were conducted in the CuSO$_4$ + H$_2$SO$_4$ solution containing 0.2 mM KCl.

### 3.2. MED and MEE Procedures

The electrochemical cell was placed at the bore center in a cryocooled superconducting magnet (Sumitomo Heavy Industries Ltd. Tokyo, Japan), which can generate magnetic fields of up to 5 T in a 220 mm bore. An imposed magnetic field **B** was parallel (+B) or antiparallel (−B) to the ionic current **i** and perpendicular to the working electrode surface. The films prepared in a magnetic field of +5 T or −5 T are termed the +5T-film or the −5T-film, respectively. The schematic configuration of the MED and MEE experiments was shown in our previous papers [4,14]. The temperature within the magnet bore was controlled at 25 °C by circulating thermo-regulated water.



*3.3. Estimation of Surface Chirality*

To examine the surface chirality of MED and MEE films, the films were used as electrodes for the voltammetric measurements of alanine enantiomers after the pre-treatment of a potential sweep (−0.3–0.3 V) in a 0.1 M NaOH aqueous solution [14]. The voltammograms of alanine were measured in an aqueous solution of 20 mM alanine + 0.1 M NaOH with a linear potential sweep rate of 10 mV s$^{-1}$ in the absence of a magnetic field. The values of *ee* ratios were estimated from experiments conducted several times at each MED or MEE condition.

## 4. Conclusions

We have demonstrated that the MED and MEE of Cu films in relatively low magnetic fields of 2~3 T induce the breaking of odd chirality for the magnetic field polarity. This result indicates that the fluctuation in the micro-MHD vortices is responsible for the broken odd chirality. On the other hand, the random fluctuation caused by the superimposed effects of the low magnetic fields and the specific adsorption of chloride ions leads to the disappearance of surface chirality. These facts imply that the ordered fluctuation in the micro-MHD vortices could be crucial for the breaking of odd chirality.

**Author Contributions:** I.M. and R.A. conceived and designed the experiments; I.M. and K.T. performed the experiments; K.T. contributed superconducting magnet tools; I.M. wrote the paper. All authors have read and agreed to the published version of the manuscript.

**Funding:** This research was funded by JSPS KAKENHI Grant-in-Aid for Scientific Research (C) No. 19K05230.

**Acknowledgments:** The authors thank the staff members of the High Field Laboratory for Superconducting Materials of IMR Tohoku University for the use of the cryocooled superconducting magnet.

**Conflicts of Interest:** The authors declare no conflict of interest.

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
