# Peer review of "Fluctuation Effects of Magnetohydrodynamic Micro-Vortices on Odd Chirality in Magnetoelectrolysis"

_magnetochemistry, doi:10.3390/magnetochemistry6030043_

Round 1

Reviewer 1 Report

The article entitled “Fluctuation Effects of Magnetohydrodynamic Micro-Vortices on Odd Chirality in Magnetoelectrolysis” written by I. Mogi, R. Aogaki, and K. Takahashi describes the results of the studies of the influence of the magnetic field strength on the chirality-dependent response of alanine measured by means of the copper electrode. The report is clearly written and the presented results are interesting, however, the manuscript requires a revision in order to be recommended for publication in Magnetochemistry.

In particular, it would be beneficial for the reader if the authors could include some additional background information since the work was built on the several previous papers published already by the authors.

The last sentence of the abstract reads: “These results imply that the fluctuation … is responsible for the breaking of odd chirality.” As this is the final conclusion of the paper, the broaden discussion why the fluctuation of the self-organized state is the reason of the braking of odd chirality seems to be very appropriate. The authors argue that the results are in agreement with their model but this is an insufficient justification since in their study they do not observe directly the surface of the electrode and one may put forward other hypothesizes explaining  their findings.

On the third page of the manuscript lines 91-93 require some explanation. Do the authors claim that in their case the kinetic and mass transfers are responsible for the sign of the quantity ee? Please notice that the term “chiral sign” should be defined in the text. It is unclear why the positive sign of ee is observed at +5 T. That is, how the absolute configuration of the molecule (L or D) is connected to the positive or negative sign of the ratio ee? This does not follow from the manuscript. Moreover, in Figures 3 and 4 the ee ratio changes sign with the current i. A brief physicochemical explanation of this phenomenon is missing in the manuscript. Please also notice that the red guide-for-eye dotted lines in Figure 3 c and d do not follow the experimental  points (there is no maximum).

Information of the estimation of experimental uncertainty should also be included. For instance, the authors could repeat their experiment for two opposite directions of the magnetic field for the racemic mixture of alanine and then to find how strongly the difference between currents recorded for each magnet configuration deviates from zero.

The authors observed that the breaking of odd chirality appears somewhere between 2.5 T and 3.0 T. Presenting data in a quantitative manner would enrich the paper, that is the authors may choose a quantity which describes how the relation ee(B)-ee(-B) changes with the strength of the magnetic field, for instance for the difference between ee(B) and ee(-B) averaged over currents.  This could give a clear picture which kind of the transition is observed in the experiment, e.g. the linear dependence up to a point where the odd chirality vanishes or the phase-like transition. This could guide the authors toward clarification of the model of the studied system.

Author Response

Answer to Referee 1

I would like appreciate valuable suggestions to our manuscript. I have revised the manuscript taking account of the suggestions. The followings are the answers to each suggestion.

Suggestion 1:  In particular, it would be beneficial for the reader if the authors could include some additional background information since the work was built on the several previous papers published already by the authors.

Answer: The background of this research was mentioned in the section of “Introduction” at lines 29-58 with reference papers. This include the model of self-organized MHD state and the experimental results of MED, MEE and specific adsorption effects. Thus, I think, this is enough volume as background.

Suggestion 2: The last sentence of the abstract reads: “These results imply that the fluctuation ... is responsible for the breaking of odd chirality.” As this is the final conclusion of the paper, the broaden discussion why the fluctuation of the self- organized state is the reason of the braking of odd chirality seems to be very appropriate. The authors argue that the results are in agreement with their model but this is an insufficient justification since in their study they do not observe directly the surface of the electrode and one may put forward other hypothesizes explaining their findings.

Answer: The experimental results of this paper strongly suggest this conclusion. But, generally, one can expect the disappearance of chirality in the fluctuation condition. Actually, when the fluctuation is too random, the surface chirality disappears as shown in Figure 7 (the superimposed effects of low fields and specific adsorption). However, the chirality exists in 2T- and 3T-films. Hence, we propose the ordered fluctuation state, as mentioned in the section 2.3 lines 167-169. But, the mechanism how the ordered fluctuation breaks the odd chirality has not yet be elucidated at the present stage.

As for the direct surface observation, Ref.10 reported the existence of a lot of screw dislocations on the surfaces of electrodeposited Cu films by the AFM measurements. The center of screw dislocation can be a chiral site, and the micro-MHD vortices could contribute to the screw dislocation growth. We have tried to observe such chiral site by means of AFM and SEM measurements, but the observation of molecular-level chiral site was very difficult.

Suggestion 3: On the third page of the manuscript lines 91-93 require some explanation. Do the authors claim that in their case the kinetic and mass transfers are responsible for the sign of the quantity ee? Please notice that the term “chiral sign” should be defined in the text. It is unclear why the positive sign of ee is observed at +5 T. That is, how the absolute configuration of the molecule (L or D) is connected to the positive or negative sign of the ratio ee? This does not follow from the manuscript. Moreover, in Figures 3 and 4 the ee ratio changes sign with the current i. A brief physicochemical explanation of this phenomenon is missing in the manuscript. Please also notice that the red guide-for-eye dotted lines in Figure 3 c and d do not follow the experimental points (there is no maximum).

Answer: The term “chiral sign” was corrected to “chirality”. The line 91-93 mentioned that the chirality L or D depends on the deposition current in the 5 T MED. The explanation for this result was reported in our previous paper Ref.16, and this explanation is beyond the issues of this paper. Thus, we mentioned it briefly and focused the attention on the magnetic field dependence of chirality.   

As for the current dependence of ee ratios in Figure 4, it is difficult to explain the current dependence of chirality especially in the presence of the fluctuation of micro-MHD vortices. This is a future issue.

The guide for eyes of red line in Figure 3c and 3d was corrected.

Suggestion 4: Information of the estimation of experimental uncertainty should also be included. For instance, the authors could repeat their experiment for two opposite directions of the magnetic field for the racemic mixture of alanine and then to find how strongly the difference between currents recorded for each magnet configuration deviates from zero.

Answer: Most experimental uncertainty is in the estimation of ee values. We conducted several times experiments at the same condition of MED and MEE. This is added to section 3.3 lines 210-211.

Suggestion 5: The authors observed that the breaking of odd chirality appears somewhere between 2.5 T and 3.0 T. Presenting data in a quantitative manner would enrich the paper, that is the authors may choose a quantity which describes how the relation ee(B)-ee(-B) changes with the strength of the magnetic field, for instance for the difference between ee(B) and ee(-B) averaged over currents. This could give a clear picture which kind of the transition is observed in the experiment, e.g. the linear dependence up to a point where the odd chirality vanishes or the phase-like transition. This could guide the authors toward clarification of the model of the studied system.

Answer: The relation ee(B)-ee(-B) vs B is very interesting, and such a relation might lead to a new point of view. However, to make such relation, it is necessary to make a lot of experiments for precise points of magnetic fields and deposition currents. Thus, we will try this in the next study.

Reviewer 2 Report

The authors experimentally observe that the previously reported electrochemical enantioselectivity of copper films, electrochemically deposited or -etched in high magnetic fields, and odd in the magnetic field,  becomes even in magnetic field for fields below 3 T. They attribute this change to fluctuations in the self-organized state of micro-MHD vortices.

The experiment is clearly described and the observations are convincing. This work could have a wide-ranging impact on many fields, ranging from chemical technology to the homochirality of life, and therefore merits to be published. I feel however that important elements are missing, and that the proposed explanation is not fully convincing. Whereas the high field odd magnetic field dependence is expected, the low field even magnetic field dependence of the electrochemical enantioselectivity is surprising, and seems even symmetry-forbidden. The fluctuation of MHD vortices could indeed make the odd dependence disappear at low field, but I fail to see how this would lead to an even field dependence.  One would rather expect all enantioselectivity to disappear at low field.  Immediately the question arises what  happens at zero field when no MHD vortices exist,  a question the authors do not address. If the enantioselectivity is observed at zero field,  this would either be a sign of a very fundamental symmetry breaking, or an experimental artifact in the setup. If the enantioselectivity disappears at zero field, the authors will have to explain a truly even magnetic field dependence of enantioselectivity.

Author Response

Answer to Referee 2

I would like appreciate valuable suggestions to our manuscript. I have revised the manuscript taking account of the suggestions. The followings are the answers to each suggestion.

Suggestions1: Whereas the high field odd magnetic field dependence is expected, the low field even magnetic field dependence of the electrochemical enantioselectivity is surprising, and seems even symmetry-forbidden. The fluctuation of MHD vortices could indeed make the odd dependence disappear at low field, but I fail to see how this would lead to an even field dependence. One would rather expect all enantioselectivity to disappear at low field.

Answer:  This is essential and significant suggestion. As the referee mentioned, the even chirality in low magnetic fields is symmetry-forbidden. However, this is the experimental results in both MED and MEE. Our experimental results strongly suggest that symmetry-forbidden phenomena are caused by the fluctuation of micro-MHD vortices. I can understand that the readers expect the disappearance of chirality in the fluctuation condition. Actually, when the fluctuation is too random, the surface chirality disappears as shown in Figure 7 (the superimposed effects of low fields and specific adsorption). However, the chirality exists in 2T- and 3T-films. Hence, we propose the ordered fluctuation state, as mentioned in the section 2.3 lines 167-169. But, the mechanism how the ordered fluctuation induce symmetry-forbidden phenomena has not yet be elucidated at the present stage.

Suggestions2: Immediately the question arises what happens at zero field when no MHD vortices exist, a question the authors do not address. If the enantioselectivity is observed at zero field, this would either be a sign of a very fundamental symmetry breaking, or an experimental artifact in the setup. If the enantioselectivity disappears at zero field, the authors will have to explain a truly even magnetic field dependence of enantioselectivity.

Answer: We have measured the ee ratios of 0T- and 1T-films of MED and MEE. Both films show no chiral behavior. The chiral surface cannot be formed when the micro-MHD vortices are absent or not fully developed. This is added in the text lines 124-125 and 145-146. The mechanism for the even chirality is a future issue.

Reviewer 3 Report

The paper "Fluctuation Effects of Magnetohydrodynamic Micro Vortices on Odd Chirality in Magnetoelectrolysis" by Mogi I. et. al presents the magnetic field dependence of copper chiral surfaces formation, investigated by MED and MEE.

The paper is well written and results are well supported by experimental data.

The topic in my opinion is really interesting and intriguing, so for these reasons I recommend this paper for the publication in Magnetochemistry Journal.

Author Response

Answer to Referee 3

We would like appreciate good evaluation and understanding for our manuscript.

Round 2

Reviewer 2 Report

The authors have clarified in the manuscript that for 0 and 1 T, the enantio-selectivity disappears, and in their accompanying letter agree that the even magnetic field dependence at 2 and 3 T is symmetry forbidden. However they do not state this highly remarkable aspect in the manuscript.  I feel they should point this out, or provide another possible explanation for the observation, the ‘fluctuation of MHD vortices’, not being convincing. For instance, a magnetic levitation force, proportional to gradient(B sup 2), could give an even magnetic field dependence.

Author Response

Answer to Referee 2

I would like appreciate valuable suggestions to our manuscript. I have revised the manuscript taking account of the suggestion. The following is the answers to the suggestion.

Suggestions1: The authors have clarified in the manuscript that for 0 and 1 T, the enantio-selectivity disappears, and in their accompanying letter agree that the even magnetic field dependence at 2 and 3 T is symmetry forbidden. However they do not state this highly remarkable aspect in the manuscript. I feel they should point this out, or provide another possible explanation for the observation, the ‘fluctuation of MHD vortices’, not being convincing. For instance, a magnetic levitation force, proportional to gradient(B sup 2), could give an even magnetic field dependence.

Answer:  I added the following sentences to the section2.1 lines 121-125.

“The even chirality is symmetry-forbidden in the MHD model, and it can be allowed in the gradient force fields. However, the electrochemical cell was placed in the homogeneous magnetic fields at the center of magnet bore, thereby there is no gradient force. These facts imply that the breaking of odd chirality could take place in low magnetic fields such that the self-organized MHD state could not be formed rigidly.”

Round 3

Reviewer 2 Report

The authors have introduced a minimal remark on the forbiddenness of an even magnetic field dependence of the enantioselectivity. With that I judge the manuscript acceptable for publication.